# A Proteome-Wide Effect of PHF8 Knockdown on Cortical Neurons Shows Downregulation of Parkinson’s Disease-Associated Protein Alpha-Synuclein and Its Interactors

**DOI:** 10.3390/biomedicines11020486

**Published:** 2023-02-08

**Authors:** Nicodemus E. Oey, Lei Zhou, Christine Hui Shan Chan, Antonius M. J. VanDongen, Eng King Tan

**Affiliations:** 1Rehabilitation Medicine, Singhealth Residency, Singapore 169608, Singapore; 2School of Optometry, Department of Applied Biology and Chemical Technology, Research Centre for SHARP Vision (RCSV), The Hong Kong Polytechnic University, Hong Kong 999077, China; 3National Neuroscience Institute, 20 College Road, Singapore 308433, Singapore; 4Department of Pharmacology and Cancer Biology, Duke University, Durham, NC 27708, USA

**Keywords:** proteomics, synaptic plasticity, neurodegeneration, Parkinson’s disease, alpha synuclein, histone demethylase PHF8

## Abstract

Synaptic dysfunction may underlie the pathophysiology of Parkinson’s disease (PD), a presently incurable condition characterized by motor and cognitive symptoms. Here, we used quantitative proteomics to study the role of PHD Finger Protein 8 (PHF8), a histone demethylating enzyme found to be mutated in X-linked intellectual disability and identified as a genetic marker of PD, in regulating the expression of PD-related synaptic plasticity proteins. Amongst the list of proteins found to be affected by PHF8 knockdown were Parkinson’s-disease-associated SNCA (alpha synuclein) and PD-linked genes DNAJC6 (auxilin), SYNJ1 (synaptojanin 1), and the PD risk gene SH3GL2 (endophilin A1). Findings in this study show that depletion of PHF8 in cortical neurons affects the activity-induced expression of proteins involved in synaptic plasticity, synaptic structure, vesicular release and membrane trafficking, spanning the spectrum of pre-synaptic and post-synaptic transmission. Given that the depletion of even a single chromatin-modifying enzyme can affect synaptic protein expression in such a concerted manner, more in-depth studies will be needed to show whether such a mechanism can be exploited as a potential disease-modifying therapeutic drug target in PD.

## 1. Introduction

Parkinson’s disease is characterized by not only motor but also cognitive symptoms [1], for which there is currently no cure. The molecular bases underlying these cognitive symptoms of PD are complex and may involve deficits in both pre-synaptic and post-synaptic transmission, which are thought to occur early in the course of the disease and therefore may serve as potential therapeutic targets [2].

Single point mutations in an epigenetic modifying enzyme can be responsible for devastating consequences, including the inability to form long-term memories [3]. Here we investigate the function of such an enzyme: PHF8 is a histone demethylase that has been implicated in two conditions: the first being X-linked intellectual disability (XLMR), a condition characterized by a profound loss of memory formation, and the second being Parkinson’s disease [4]. Previously, we found evidence that PHF8 regulates the activity-induced expression of the neuronal protein ARC [5], a major regulator of synaptic function [6]. Synaptic dysfunction is indeed found in early stages of PD neuropathology and is mediated by apparent pathological elevation of Alpha-synuclein (SNCA), a PD-implicated protein that is central to the synaptic dystrophy that precedes dopaminergic cell loss [7], whose dysfunction contributes to synaptic failure [8], affirming the classification of PD as a synaptopathy [9].

In this study, we used mass-spectrometry-based proteomics combined with RNA interference (RNAi) to knock down PHF8 to quantify its effects on the levels of neuronal proteins. We quantified proteomes of primary cortical neurons cultured from Rattus norvegicus with or without PHF8 depletion after an 8 h period of increased synaptic activity. We observed that the RNAi-induced knockdown of PHF8 specifically decreased the expression of key players in synaptic plasticity such as SNCA, Calcium–calmodulin-dependent kinase II alpha (CaMKIIa), and Complexin 1 (CPLX1) as well as several important interacting proteins involved in the synaptic vesicular pool regulation, vesicle release, and membrane trafficking, thereby encompassing the full spectrum of synaptic function (Table 1). Of note, the three recently identified PD-linked genes that are involved in synaptic vesicle endocytosis, namely DNAJC6 (auxilin), SYNJ1 (synaptojanin 1), and SH3GL2 (endophilin A1) [10], were all downregulated by PHF8 knockdown (Table 1).

Whereas SNCA is a presynaptically active protein that is causally linked to Parkinson’s disease through its role in the biosynthesis, release, and reuptake of dopamine [11], CaMKIIa is an important kinase that is enriched in mammalian synapses and plays critical roles in modulating synaptic transmission, which has also been found to interact with the dopamine receptor and thus play a role in PD pathogenesis (Table 1) [2,12]. CPLX1 is a cytosolic protein that functions in synaptic vesicle exocytosis and has been shown to be increased in PD brains [13]. Interestingly, CPLX1 expression is tightly linked to SNCA [14], and its transcription has been found to be a possible biomarker for sporadic PD [15,16]. Finally, the current proteomic dataset also shows that PHF8 affects the levels of three more PD-linked synaptic proteins: DNAJC6, which is implicated in synaptic vesicle trafficking and clathrin dynamics and is genetically associated with PD [17]; its functionally linked protein SYNJ1, which is involved in synaptic vesicle recycling [18]; and yet another endosomal protein SH3GL2, which may play a role in PD pathogenesis through its regulation of synaptic vesicle endocytosis [10]. The finding that such clusters of functionally linked proteins are affected by the downregulation of a single chromatin-modifying enzyme such as PHF8 is intriguing and deserves further study to characterize its possible role in the pathophysiology and eventual treatment of PD.

## 2. Materials and Methods

### 2.1. Antibodies

Sheep polyclonal anti-alpha synuclein (ab6162 from Abcam) was used for immunostaining at 1:1000 dilution using antibody dilution buffer, which was 1× PBS containing 1% (*w*/*v*) bovine serum albumin (BSA), 5% (*v*/*v*) serum and 0.05% (*v*/*v*) Triton X-100). Incubation with primary antibody was performed for 1 h at room temperature, after which cells were washed three times in 1× PBS-Tx. Cells were then probed with 1:1000 anti-mouse secondary antibodies coupled with Alexa-Fluor 568 (Molecular Probes, Eugene, OR, USA) for 1 h at room temperature. Staining of DNA with 50 μM 40,6-diamidino-2-phenylindole (DAPI) was performed for 20 min at room temperature. Cells were mounted in FluorSave (Calbiochem, San Diego, CA, USA).

### 2.2. Primary Culture of Cortical Neurons

Cortices were dissected from E18 embryos of Sprague–Dawley rats (Rattus norvegicus), which were then subjected to the Papain Dissociation System (Worthington Biochemical Corporation, Lakewood, CA, USA). Dissociated cells were plated on poly-D-lysine-coated dishes at a plating density of 1.5 × 105/cm^2^ in neurobasal medium (Gibco, Grand Island, New York, NY, USA) supplemented with 10% (*v*/*v*) fetal bovine serum (FBS), 1% (*v*/*v*) penicillin–streptomycin (P/S, Gibco, Grand Island, New York, NY, USA), and 2% (*v*/*v*) B27 supplement (Gibco, Grand Island, New York, NY, USA) for 2 h. Medium was changed on days in vitro (DIV) 5. Subsequently, medium was changed every three to four days. All experiments were carried out on DIV 21 as previously described [5].

### 2.3. Pharmacological Stimulation of Neural Network Activity

Primary rat cortical neuronal cultures were treated with a combination of 100 μM 4-aminopyridine (4AP), 50 μM bicuculline, and 50 μM forskolin (hereafter abbreviated as 4BF) for 8 h to induce neural network activity as previously described [4]. This protocol has been previously reported to induce pharmacological LTP [66,67].

### 2.4. Cell Lysate Preparation and RNA/Protein Extraction

Following 4BF stimulation, neuronal cultures were washed gently with phosphate-buffered saline (PBS). Cells were gently scraped off and harvested in an Eppendorf tube. Cells were spun down at 10,000× *g* for 5 min at 4 °C to obtain the cell pellet. Total protein was isolated using an RNA/protein extraction kit (Macherey-Nagel, Düren, North Rhine–Westphalia, Germany) as specified by the manufacturer. A BCA kit (Pierce, Rockford, IL, USA) was used to measure the concentration of proteins. A total of 30 μg of each protein sample was used for subsequent mass spectrometry experiments. For qRT-PCR studies, 4BF-treated neuronal cells were washed, scraped, and spun down as above. RNA samples were obtained from the cell pellet using the RNA-protein extraction kit as specified by the manufacturer (Macherey-Nagel, Düren, North Rhine–Westphalia). cDNA was synthesized and subsequently purified using a spin column (Qiagen), then eluted into 30 μL volumes; thereafter, 2 μL was used in qRT-PCR employing SYBR Green using primers against the transcriptional start site (TSS) of known activity-regulated neuronal genes such as Arc (NCBI Gene ID: 54323), BDNF (NCBI Gene ID: 24225), Fos (NCBI Gene ID: 314322), and control genes including Rpl19 (NCBI Gene ID: 81767) and GAPDH (NCBI Gene ID: 24383). The primers used for PHF8 qRT-PCR were CCTAAAGCCCGTGTGACT and GGCGCGGCTGTTCTACCT. Statistical analyses were performed using Student’s two-tailed *t*-test with a *p*-value < 0.05 being considered significant.

### 2.5. Isobaric Tag for Relative and Absolute Quantification of Proteins (iTRAQ)

Following reduction and alkylation, proteins collected from primary cortical neurons were trypsinized overnight at 37 °C. Peptides were dried and resuspended in mass-spectrometry-compatible buffer and then labeled with the 8-plex iTRAQ labeling reagent (Applied Biosystems, Foster City, CA, USA). Labeled samples were combined and analyzed with one-dimensional nanoLC-MS/MS (Dionex UltiMate 3000 nanoLC system coupled with AB Sciex TripleTOF 5600 system) for protein identification. The IPI human protein database (version 3.77) was searched using ProteinPilot (version 4.5, AB Sciex, Framingham, MA, USA) and the identified hits were analyzed using DAVID (http://david.abcc.ncifcrf.gov; accessed on 11 January 2023) for gene ontology annotation [68]. The mass spectrometry proteomics data were deposited to the ProteomeXchange Consortium via the PRIDE partner repository with the dataset identifier PXD036335 [69]. 

### 2.6. Immunofluorescence

Primary cortical neuronal cells transfected with expression vectors or shRNAs as indicated were pre-extracted with 100% methanol or directly fixed for 5 min with 4% formaldehyde in sucrose buffer, then permeabilized for 2 min with 0.1% Triton X-100 if there was no pre-extraction. After three rinses with PBS/0.1% Triton X-100, blocking solution (10% BSA and goat serum in PBS, pH 7.4) was applied for 30 min and primary antibodies were added in blocking buffer for 1 h at room temperature. After three 5 min washes with PBS/0.1% Triton X-100, cells were incubated with secondary antibodies conjugated with fluorescent dyes (Alexa Fluors, Invitrogen) for 1 h, washed again with PBS/0.1% Triton X-100, and mounted in 97% thiodiethanol in PBS (Sigma-Aldrich, St. Louis, MO, USA). Images were recorded on a Nikon-Ti microscope (Nikon, Tokyo, Japan) equipped with an Andor camera (Andor, Belfast, Ireland) at 1 × 1 binning and a 60× objective (Nikon). *Z*-stacks (0.2 μm sections) were deconvolved using AutoQuant (Nikon NIS Elements) and projected for maximum intensity. Image intensities for each antibody were scaled identically. 

### 2.7. Widefield Imaging Microscopy

Fluorescence images were obtained using a motorized inverted wide-field epifluorescence microscope (Nikon Eclipse Ti-E) using the 20× objective lens. Motorized excitation and emission filter wheels (Ludl electronics, Hawthorne, NY, USA) fitted with a DAPI/CFP/YFP/DsRed quad filter set (#86010, Chroma, Rockingham, VT, USA) were used together with filter cubes for DAPI, CFP, YFP, and TxRed (Chroma) to select specific fluorescence signals. Z-stacks were obtained spanning the entire nucleus and out-of-focus fluorescence was removed using the AutoQuant deconvolution algorithm (Media Cybernetics). Images were digitized using a cooled EM-CCD camera (iXon EM+ 885, Andor, Belfast, Northern Ireland). Image acquisition was performed using NIS Elements AR 4.2 software (Nikon). NIS Elements Binary and ROI Analysis tools were used to segment nuclei based on DAPI signal intensity.

### 2.8. Overexpression of PHF8-YFP and Knockdown of PHF8 Levels Using shRNA and siRNA Transfection

For the short-hairpin RNA (shRNA) experiments, a combination of two shRNA plasmids targeting the PHF8 sequence were used: RLGH-EN02366 and RLGH-EU01979 (Transomic Technologies, Huntsville, AL, USA). Neuronal cultures (DIV21) were transfected with either the shRNA plasmids or a plasmid containing PHF8 fused to Yellow Fluorescent Protein (YFP) using the Lipofectamine 2000 reagent (Invitrogen, Carlsbad, CA, USA) according to the manufacturer’s protocol. For the overexpression experiments, DNA containing PHF8-YFP construct that was previously created [4] was added to Lipofectamine at a ratio of 1:1. The Lipofectamine:DNA complex was incubated at room temperature for 20 min before being added to the cells. The complex was added dropwise such that it was evenly distributed on the cell culture. Culture medium was added after 20 min and experiments were performed on DIV21.

For the small interfering RNA (siRNA) experiments, a combination of three unique 27 mer siRNA duplexes targeting the PHF8 sequence were used (Locus ID 317425, Origene, Rockville, MD, USA) and transfected into primary cortical neuronal cells at DIV19 using the LipofectamineRNAiMax reagent according to the manufacturer’s protocol (Invitrogen, Carlsbad, CA, USA). After transfection, neuronal medium was changed, and neurons were allowed to equilibrate prior to sample collection at DIV21 for mass spectrometry analyses.

## 3. Results

### 3.1. Relative Quantitation of the Activity-Regulated Cortical Neuronal Proteome

We performed the quantitative proteomics method of isobaric tag for relative and absolute quantitation (iTRAQ), which has been validated for protein quantitation in neural tissues [70,71]. Using iTRAQ, we analyzed primary cultured cortical neurons from Rattus norvegicus. After combining 2 biological replicates, we identified 2678 unique proteins though multidimensional protein identification (MudPIT), out of which 2656 proteins were confirmed with more than one peptide (Figure 1, Appendix A). 

### 3.2. RNA Interference Specifically Depleted PHF8 in Primary Rat Cortical Neurons

We used a specific siRNA directed against rat PHF8 and scrambled siRNA as a control to transfect primary neuronal cultures. In order to that validate PHF8 knockdown was successful at the transcript level, neurons were subjected to RT-PCR 3 weeks after at DIV 21, after an 8 h stimulation protocol with 100 μM 4-Aminopyridine, 50 μM Bicuculline, and 50 μM Forskolin. Transcript levels of control genes such as the ribosomal protein Rpl19, the housekeeping gene GAPDH, and another transcriptional regulator p300 served as the negative controls (Figure 2). 

### 3.3. Validation of Alpha-Synuclein as a Synaptic Protein Regulated by PHF8

As alpha-synuclein (SNCA), a critical protein in Parkinson’s disease pathogenesis, turned out to be amongst the main proteins identified via proteomics to be downregulated by PHF8 depletion (Table 1). In order to validate this iTRAQ finding, we transfected neurons with PHF8 to overexpress it, then contrasted these neurons against neurons transfected with a specific shRNA against PHF8 (Figure 3). 

### 3.4. Pathway Analysis Using DAVID Reveals Specific Downregulation of Proteins in PHF8 Knockdown That Are Involved in Synaptic Function

Out of the 2656 unique proteins quantified using MudPIT (Appendix A), we manually curated 41 proteins that were selected based on their physiological function and role in human diseases (Table 1). Amongst the proteins that were downregulated by PHF8 depletion in an activity-dependent manner, at least 33 were involved in synaptic function (Table 1). A search using the functional enrichment analysis tool DAVID led to the discovery of top terms including “synaptic vesicle endocytosis”, “modulation of synaptic transmission”, and “synapse organization” (Table 2) [68].

## 4. Discussion

### 4.1. The Potential Role of PHF8 in Synaptic Plasticity and PD Pathogenesis

Synaptic plasticity in the form of long-term potentiation (LTP) and long-term depression (LTD) may be dysregulated in Parkinson’s disease even before motor symptoms start to manifest. In this mass-spectrometry-based proteomics study using primary rat cortical neurons, 2605 unique proteins that were identified as differentially regulated by PHF8, out of which the top 41 proteins are listed in Table 1, had functions in synaptic transmission (Table 2). As this was the first attempt to assess the role of PHF8 in an in vitro model of PD pathogenesis, we were not able to compare our results against published data in the literature. Nonetheless, the protein targets curated in the current dataset can be crossmatched with published PHF8 genetic targets obtained by various groups via ChipSeq (Appendix A [72]). 

We previously found that PHF8 is required for the transcription of Arc, the master regulator of synaptic plasticity [5]. Now, using mass spectrometry coupled with the isobaric tag for relative and absolute quantitation (iTRAQ) method, we quantified proteome-wide effects of PHF8 depletion in primary cortical neurons stimulated using a combination of three pharmacological agents to induce chemical LTP [66,67] and report that PHF8 may play a regulatory role in the activity-induced expression of neuronal proteins such as Alpha-synuclein and Calmodulin Kinase II Alpha, which in turn play critical roles in the function of neuronal synapses and the pathophysiology of PD. Pathway analysis using DAVID allowed for the identification of groups of proteins affected by PHF8 with distinct functions: with respect to LTP, the Calmodulin Kinase II Alpha-Calmodulin-dependent Kinase 2 axis was downregulated by PHF8 depletion (Table 2, Appendix A). Multiple synaptic transmission pathways that have been implicated in PD pathogenesis are seen to converge as NSF, AMPH, EIF4E, and CHN1 are all downregulated by PHF8 (Table 2). Specific to clathrin-mediated endocytosis of synaptic vesicles, the pre-synaptic protein auxilin (DNAJC6), synaptojanin-1 (SYNJ1), and endophilin A1 (SH3GL2) [10] were also affected by PHF8 downregulation. These subgroups of PHF8-regulated synaptic proteins show that the knockdown of a single epigenetic enzyme can have a highly concerted effect on synaptic function. 

### 4.2. Knockdown of PHF8 Reduces Levels of Alpha-Synuclein

This proteomics study revealed that levels of SNCA reduced significantly with PHF8 knockdown, which we attempted to validate using immunofluorescence microscopy (Figure 3). The effective reduction of SNCA levels by PHF8 knockdown raises the possibility of potentially using this mechanism as a potential therapeutic target by reducing the function of PHF8, either via gene therapy [73] or through pharmacological means, since PHF8 is an enzyme.

### 4.3. Knockdown of PHF8 Reduces the Levels of Other Synaptic Plasticity-Related Proteins and SNCA Interactors

In addition to CamKIIa, amongst the other synaptic proteins that were differentially regulated by PHF8 in primary cortical neurons, we highlight CPLX1, which is involved in stimulus-dependent control of synaptic vesicle exocytosis and is a known interactor of SNCA [14]. Another interesting target from the current proteomics dataset is Rabphilin-3A (RPH3A), which is implicated in levodopa-induced dyskinesia [74] and is dysregulated in PD. Finally, CHN1, which was identified in a gene network in PD and AD [75], was also upregulated by PHF8 knockdown in our dataset. SNCA is one of many possible target genes of PHF8, according to the publicly available ChIP-Atlas dataset [19], supplementing the current proteomic-level observation (Appendix A) [72]. 

In summary, the concerted downregulation of not just Alpha-synuclein, but also many of its interactors, which have various roles in synaptopathy by PHF8 knockdown, deserves further study. Further validation of the role of a single epigenetic modifying enzyme such as PHF8 in a potential disease-modifying therapy through modulation of synaptic function should be considered.

## Figures and Tables

**Figure 1 biomedicines-11-00486-f001:**
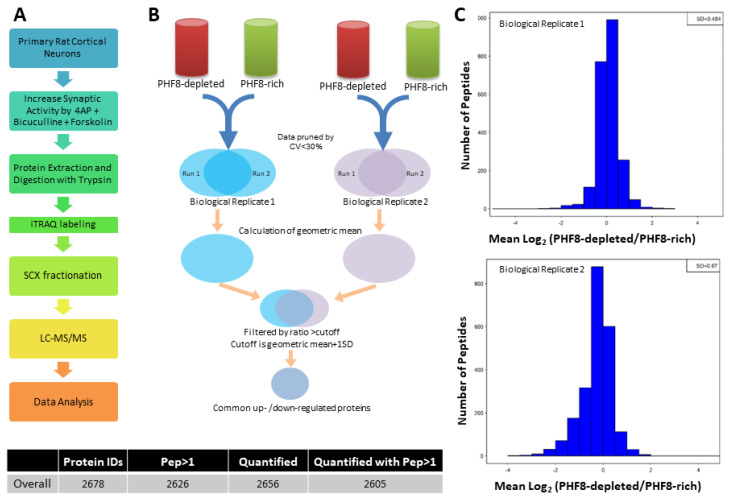
Schematic representation of the proteomic study protocol and the geometric means of biological replicates used in the study. (**A**) Workflow of proteomic experiment. (**B**) Schematic of the iTRAQ quantitative proteomics protocol where two biological replicates are run through the mass spectrometer in duplicate (Run 1 and Run 2), which are then combined. (**C**) Normogram of the number of peptides against the mean log2 frequency distribution.

**Figure 2 biomedicines-11-00486-f002:**
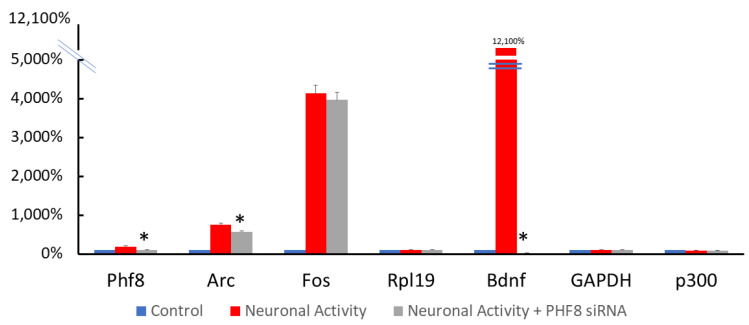
Knockdown of PHF8 transcription via RNA interference using small interfering RNA (siRNA)—values of 2^−delta(delta CT)^ normalized to a ribosomal gene Rpl19 in cortical neurons, which are quiescent (control) compared to neurons subjected to 3 h of 4AP + Bicuculline + Forskolin (4BF) with and without PHF8 siRNA, showing effective knockdown of the expression of PHF8 at 3 weeks (DIV 21) on neuronal and glial markers. Data are represented as mean ± SEM. All differences that are significant with a * *p*-value of <0.05 (Student’s two-tailed *t*-test) were marked with an asterisk.

**Figure 3 biomedicines-11-00486-f003:**
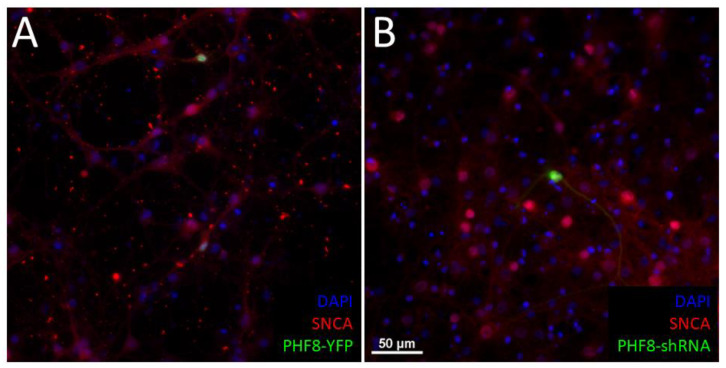
(**A**) A representative image of DIV 21 cortical neurons transfected with PHF8-YFP (in green) immunostained with an antibody against Alpha-synuclein (SNCA, in red) showing that the overexpression of PHF8 does not seem to affect SNCA levels (the level of red signal did not change compared to other neurons that are not transfected with green and due to the mixture of red and green the cell has turned slightly yellow). Scale bar = 50 um. (**B**) Transfection of a shRNA plasmid against PHF8 results in a qualitative reduction of SNCA (the green neuron does not turn yellow; not quantified).

**Table 1 biomedicines-11-00486-t001:** A manually curated shortlist of the top-ranked proteins that are downregulated when PHF8 is knocked down in activated neurons showing a functional enrichment of synaptic plasticity and Parkinson’s disease. The full set of hits can be found in Appendix A. The genes encoding for the proteins that are in bold are known to be potential target genes of PHF8 as per the ChIP-Atlas [19].

Proteins downregulated by PHF8 knockdown				
Common Name	ID	Ratio	CV	Role in Human Diseases
Long-Term Potentiation (LTP) and Synaptic Plasticity				
Calcium/Calmodulin Dependent Protein Kinase II Alpha	CaMKIIa	0.239	0.352	Master regulator of synaptic plasticity; may be overactive in Parkinson’s disease [12,20]
Calcium/Calmodulin-dependent Protein Kinase Kinase 2	CaMKK2	0.251	0.795	Regulates LTP in the hippocampus [21]
Growth-associated Protein 43/Neuromodulin	GAP43	0.306	0.044	Enriched in the substantia nigra; regulates axonal regeneration in Parkinson’s disease [22]
Alpha-Chimerin	CHN1	0.275	0.01	Regulates dendritic spines [23]; downregulated in Parkinson’s disease [24]
Calbindin	CALB1	0.457	0.259	Physically sequesters SNCA; mutation protects against aggregation in Parkinson’s disease [25]
Alpha-Synuclein	SNCA	0.329	0.01	First gene identified to be causative for Parkinson’s disease [26]; regulates pre-synaptic vesicle pool [27]
Synapsin 1	SYN1	0.667	0.381	Modulates neurotransmitter release, vesicular fusion and recycling [28]
Synaptic Structure				
Neural Cell Adhesion Molecule 1	NCAM1	0.256	0.05	Mediates long-term potentiation [29], serves as receptor for GDNF ligands [30]
Microtubule-Associated Protein 2	MAP2	0.348	0.052	Major component of neuronal dendrites, colocalizes with SNCA in Lewy bodies [31]
Microtubule-Associated Protein 4	MAP4	0.310	0.156	Regulates synaptic vesicles along microtubular tracks, implicated in Alzheimer’s disease [32]
Microtubule-associated protein 1B	MAP1B	0.661	0.169	Controls dendritic structure [33], implicated in intellectual disability, schizophrenia, and neurodegeneration [34]
Brain Acid Soluble Protein 1	BASP1	0.468	0.006	Regulates morphology of neuronal membranes [35]
Ankyrin 2	ANK2	0.375	0.454	Regulates synaptic stability, implicated in autism spectrum disorder [36]
L1 Cell Adhesion Molecule	L1CAM	0.472	0.527	Organizes the neuronal ankryin–spectrin interactions, implicated in MASA syndrome characterized by intellectual disability [37]
Signal Regulatory Protein α	SIRPA	0.515	0.64	Organizes pre-synaptic vesicle clusters [38], underlies synaptic maturation [39]
A-kinase anchoring protein 5	AKAP5	0.53	0.521	Scaffolding for PSD95 and SAP97, implicated in schizophrenia [40]
Synaptic Vesicle Endocytosis/Release				
Synaptojanin 1	SYNJ1	0.525	0.162	Mutation in the SYNJ1 gene associated with autosomal recessive, early-onset Parkinsonism
Endophilin A1	SH3GL2	0.685	0.026	Together with SNCA, drives membrane bending and clathrin pit formation [41]
Auxilin	DNAJC6	0.182	0.307	A major presynaptic endocytic protein linked to PD [42]
Complexin-1/Synaphin	CPLX1	0.435	0.046	Interacts with SNCA; critical in the regulation of neurotransmitter release [43]
Rabphilin 3A	RPH3A	0.299	0.01	Interacts with SNCA; involved in neurotransmitter release in Parkinson’s disease [44]
Rab-like protein 6	RABL6	0.388	0.01	Analogue of Rab3, involved in neurotransmitter release [45]
ZW10-interacting protein	ZWINT	0.487	0.855	Binds to Rab3C, regulates synaptic vesicle release [46]
Vesicle transport through interaction with T-SNAREs 1A	VTI1A	0.525	0.021	Mediates spontaneous neurotransmitter release [47], implicated in Alzheimer’s disease [48]
Syntaxin 7	STX7	0.611	0.045	Downregulated in aging mice in a Tau-dependent manner [49]
Synaptic Translation				
Eukaryotic Initiation Factor 4E	EIF4E	0.637	0.032	Interacts with Parkin; regulates synaptic protein translation [50]
Eukaryotic Elongation Factor, Selenocysteine-TRNA-Specific	EEFSEC	0.402	0.045	Regulates selenium protein synthesis [51]
Synaptic Membrane Recycling/Trafficking				
Ras-related protein 2B	RAB2B	0.603	0.287	Involved in Golgi body fragmentation; downregulated in aging [52]
N-ethylmaleimide sensitive fusion	NSF	0.573	0.097	Interacts with SNCA; an independent risk locus for Parkinson’s disease; downregulated in Parkinson’s disease [53]
Clathrin, light-chain	CLTA	0.446	0.219	Directly binds to LRRK2, regulates dendritic spine morphology [54]
Phosphatidylinositide phosphatase SAC2	INPP5F	0.525	0.045	Involved in clathrin-mediated endocytosis, identified in multiple GWAS studies as a validated independent risk locus for Parkinson’s disease [55]
Soluble N-ethylmaleimide-sensitive factor (NSF) attachment protein 29	SNAP29	0.335	1.143	Negative regulator of neurotransmitter release [56]
Clathrin Coat Assembly Protein AP180	SNAP91	0.581	0.078	Mediates clathrin-mediated vesicle recycling [57]
Ubiquitin-conjugating enzyme E2 variant 2	UBE2V2	0.628	0.277	Regulates glutamate receptor trafficking [58]
Amphiphysin	AMPH	0.631	0.345	Links Dynamin with Clathrin to mediate endocytosis, mediated in SCNA-related neurodegeneration [59]
Axonal Proteins				
Dedicator Of Cytokinesis 7	DOCK7	0.394	0.034	Regulates axonal polarity [60], implicated in epileptic encephalopathy [61]
Netrin G2	NTNG2	0.616	0.175	Regulates axonal pathfinding [62], implicated in schizophrenia [63]
Proteins with Nuclear Roles/Transcriptional Modulation				
DNA Methyltransferase 1	DNMT1	0.353	0.032	Critical DNA methyltransferase for activity-regulated synaptic function [64]
DNA Methyltransferase 3A	DNMT3A	0.752	0.328	Critical DNA methyltransferase for activity-regulated synaptic function [64]
Serine/Arginine-Rich Splicing Factor 11	SRSF11	0.51	0.173	Regulates Tau gene splicing [65]
RNA binding motif protein 17	RBM17	0.649	0.189	RNA metabolism and splicing, implicated in spinocerebellar ataxia

**Table 2 biomedicines-11-00486-t002:** Pathway analysis using DAVID showing the biological functions (GOTERM_BP_DIRECT) most affected by PHF8 knockdown, ranked by gene count, in primary cortical neurons. The full list containing all hits can be obtained in Appendix A.

GO TERM	Biological Process	%	*p*-Value	Fold Enrichment	Benjamini	FDR
GO:0006886	Intracellular protein transport	4.13257	7.81 × 10^−25^	2.964188	4.93 × 10^−21^	4.74 × 10^−21^
GO:0006457	Protein folding	2.536825	1.04 × 10^−22^	3.917003	3.29 × 10^−19^	3.16 × 10^−19^
GO:0015031	Protein transport	4.13257	8.30 × 10^−22^	2.731703	1.75 × 10^−18^	1.68 × 10^−18^
GO:0007420	Brain development	4.500818	4.41 × 10^−18^	2.346359	6.96 × 10^−15^	6.69 × 10^−15^
GO:0007409	Axonogenesis	2.12766	6.74 × 10^−18^	3.710042	8.51 × 10^−15^	8.19 × 10^−15^
GO:0006888	ER to Golgi vesicle transport	2.00491	6.12 × 10^−16^	3.557335	6.44 × 10^−13^	6.19 × 10^−13^
GO:0016192	Vesicle-mediated transport	2.782324	3.34 × 10^−15^	2.772339	3.01 × 10^−12^	2.90 × 10^−12^
GO:0050808	Synapse organization	1.595745	2.46 × 10^−14^	3.888839	1.94 × 10^−11^	1.87 × 10^−11^
GO:0031175	Neuron projection development	2.536825	7.92 × 10^−14^	2.758749	5.56 × 10^−11^	5.35 × 10^−11^
GO:0010976	Neuron projection regulation	2.454992	1.20 × 10^−13^	2.789746	7.57 × 10^−11^	7.28 × 10^−11^
GO:0050804	Modulation of synaptic transmission	2.00491	4.79 × 10^−13^	3.072244	2.75 × 10^−10^	2.65 × 10^−10^
GO:0000226	Microtubule cytoskeleton organization	2.045827	5.66 × 10^−13^	3.020528	2.98 × 10^−10^	2.87 × 10^−10^
GO:0050821	Protein stabilization	2.495908	1.05 × 10^−12^	2.6432	5.08 × 10^−10^	4.89 × 10^−10^
GO:1990090	Response to nerve growth factor stimulus	1.309329	4.20 × 10^−12^	3.952835	1.89 × 10^−9^	1.82 × 10^−9^
GO:0050790	Regulation of catalytic activity	4.255319	6.06 × 10^−12^	1.987829	2.55 × 10^−9^	2.46 × 10^−9^
GO:0006099	Tricarboxylic acid cycle	0.859247	9.07 × 10^−12^	5.60649	3.58 × 10^−9^	3.45 × 10^−9^
GO:0048488	Synaptic vesicle endocytosis	1.104746	7.36 × 10^−11^	4.138124	2.73 × 10^−8^	2.63 × 10^−8^
GO:0007005	Mitochondrion organization	1.595745	2.05 × 10^−10^	3.045035	6.80 × 10^−8^	6.55 × 10^−8^
GO:0032981	Mitochondrial respiratory chain complex I	1.186579	2.34 × 10^−10^	3.750175	7.39 × 10^−8^	7.11 × 10^−8^
GO:0016310	Phosphorylation	2.086743	6.63 × 10^−10^	2.512432	1.99 × 10^−7^	1.92 × 10^−7^
GO:0007269	Neurotransmitter secretion	0.859247	2.81 × 10^−9^	4.456441	8.07 × 10^−7^	7.77 × 10^−7^
GO:0021987	Cerebral cortex development	1.636661	3.15 × 10^−9^	2.758749	8.65 × 10^−7^	8.32 × 10^−7^
GO:0007030	Golgi organization	1.554828	3.63 × 10^−9^	2.83331	9.54 × 10^−7^	9.18 × 10^−7^
GO:0072659	Protein localization to plasma membrane	2.00491	5.14 × 10^−9^	2.42836	1.30 × 10^−6^	1.25 × 10^−6^
GO:0061077	Chaperone-mediated protein folding	0.818331	6.82 × 10^−9^	4.473647	1.60 × 10^−6^	1.54 × 10^−6^
GO:0007612	Learning	1.186579	8.56 × 10^−9^	3.287824	1.93 × 10^−6^	1.86 × 10^−6^
GO:0021766	Hippocampus development	1.513912	2.92 × 10^−8^	2.686151	6.37 × 10^−6^	6.13 × 10^−6^
GO:0032456	Endocytic recycling	1.06383	3.47 × 10^−8^	3.362226	7.30 × 10^−6^	7.03 × 10^−6^
GO:0030036	Actin cytoskeleton organization	2.045827	4.41 × 10^−8^	2.26127	8.73 × 10^−6^	8.40 × 10^−6^

## Data Availability

The mass spectrometry proteomics data have been deposited to the ProteomeXchange Consortium via the PRIDE [69] partner repository with the dataset identifier PXD036335.

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
