# Peer review of "A Proteome-Wide Effect of PHF8 Knockdown on Cortical Neurons Shows Downregulation of Parkinson’s Disease-Associated Protein Alpha-Synuclein and Its Interactors"

_biomedicines, 2023, doi:10.3390/biomedicines11020486_

Round 1

Reviewer 1 Report

Review of a manuscript “A proteome-wide effect of PHF8 knockdown on cortical neurons shows downregulation of Parkinson’s disease-associated protein Alpha-synuclein and its interactors” by Nicodemus E. Oey and coauthors submitted to “Biomedicines”.

Parkinson’s disease is the second after Alzheimer’s disease neuro-degenerative disorder for which there is no treatment modifying the course of the illness. Therefore, research to investigate molecular and cellular mechanism of this disease is of great importance. The authors used mass spectrometry-based proteomics in combination with RNA interference (RNAi) to knock down a chromatin-modifying enzyme  histone demethylase PHF8. The aim of this study was to examine the levels of neuronal proteins with or without PHF8 depletion and to determine the effect increased synaptic activity. This is an important direction of biomedical science and the results presented in the manuscript will be interesting for the readers of “Biomedicines”.

The following corrections and additions should be made.

Abstract

Line 14: “…a chromatin-modifying enzyme found to be mutated in intellectual disability”. The authors should be more specific saying intellectual disability. What do they mean:  cognitive impairments, impaired learning and memory, loss of memory, impaired long-term potentiation?

Keywords: the authors should add alpha-synuclein and histone demethylase PHF8.

Introduction

Lines 27-28:” Parkinson’s disease is characterized by not only motor but also cognitive symptoms, for which there is currently no cure.” The authors should add a reference to a recently published review on Parkinson’s disease: ” Biomarkers in Parkinson’s Disease”. Chapter in a book, Peplow P.V., Martinez B., Gennarelli T.A. (eds) Neurodegenerative Diseases Biomarkers. 2022. Neuromethods, vol 173. pp 155-180. Humana, New York, NY. https://link.springer.com/protocol/10.1007/978-1-0716-1712-0_7

Lines 45-47: ”In this study, we used mass spectrometry-based proteomics combined with RNA interference (RNAi) to knock down PHF8 to quantify the levels of neuronal proteins in primary cortical neurons cultured from Rattus norvegicus with or without PHF8 depletion after an 8-hour period of increased synaptic activity”. This sentence is too long and overloaded with unnecessary details. It can be made more concise, for example: ”In this study, we used mass spectrometry-based proteomics combined with RNA in terference (RNAi) to knock down PHF8 and quantify its effect on the levels of neuronal proteins”.

Materials and methods.

Line 90: ”..10,000x g for 5 min at 4C to obtain the cell pellet. Total protein was isolated using … “. This should be corrected as : ”… 10,000x g for 5 min at 40 C to obtain the cell pellet. Total protein was isolated using …” Also “Materials and methods” section contains some details which may be truncated referring to the previously published methods.

Line 94:” For rt-PCR studies,…” should be written as “For RT-PCR studies,…”

Line 100 – the same correction as above.

Line 102: “2.5. iTRAQ (Isobaric Tag for Relative and Absolute Quantitation) of proteins” The term quantification may be more appropriate here.

Figure 3

Notes on the parts A and B of the should be placed in similar corners of the figures (right or left)

Author Response

Abstract

Line 14: “…a chromatin-modifying enzyme found to be mutated in intellectual disability”. The authors should be more specific saying intellectual disability. What do they mean:  cognitive impairments, impaired learning and memory, loss of memory, impaired long-term potentiation?

  • Response: we have amended the Abstract to read: Here we used quantitative proteomics to study the role of PHF8, a chromatin-modifying enzyme found to be mutated in X-linked mental retardation and identified as a genetic marker of PD, in regulating the expression of PD-related synaptic plasticity proteins, amongst which the Parkinson’s disease-associated alpha synuclein (SNCA) and PD-linked genes DNAJC6 (auxilin), SYNJ1 (synaptojanin 1), and the PD risk gene SH3GL2 (endophilin A1) were all downregulated by PHF8 knockdown.

Keywords: the authors should add alpha-synuclein and histone demethylase PHF8.

  • Response: Thank you, Reviewer, for this excellent comment. We have added these keywords to the article.

Introduction

Lines 27-28:” Parkinson’s disease is characterized by not only motor but also cognitive symptoms, for which there is currently no cure.” The authors should add a reference to a recently published review on Parkinson’s disease: ” Biomarkers in Parkinson’s Disease”. Chapter in a book, Peplow P.V., Martinez B., Gennarelli T.A. (eds) Neurodegenerative Diseases Biomarkers. 2022. Neuromethods, vol 173. pp 155-180. Humana, New York, NY. https://link.springer.com/protocol/10.1007/978-1-0716-1712-0_7

  • Response: thank you, we have added this excellent review as one of the references.

Lines 45-47: ”In this study, we used mass spectrometry-based proteomics combined with RNA interference (RNAi) to knock down PHF8 to quantify the levels of neuronal proteins in primary cortical neurons cultured from Rattus norvegicus with or without PHF8 depletion after an 8-hour period of increased synaptic activity”. This sentence is too long and overloaded with unnecessary details. It can be made more concise, for example: ”In this study, we used mass spectrometry-based proteomics combined with RNA in terference (RNAi) to knock down PHF8 and quantify its effect on the levels of neuronal proteins”.

  • Response: thank you, Reviewer. We have edited the sentence as requested to be more concise. It currently reads: In this study, we used mass spectrometry-based proteomics combined with RNA interference (RNAi) to knock down PHF8 to quantify its effects on the levels of neuronal proteins. We quantified proteomes of primary cortical neurons cultured from Rattus norvegicus with or without PHF8 depletion after an 8-hour period of increased synaptic activity.

Materials and methods.

Line 90: ”..10,000x g for 5 min at 4C to obtain the cell pellet. Total protein was isolated using … “. This should be corrected as : ”… 10,000x g for 5 min at 40 C to obtain the cell pellet. Total protein was isolated using …” Also “Materials and methods” section contains some details which may be truncated referring to the previously published methods.

  • Response: thank you, Reviewer for this comment. We have revised the sentence, and have also added more sections to the Materials and Methods.

Line 94:” For rt-PCR studies,…” should be written as “For RT-PCR studies,…”

  • Response: we have corrected all instances of rt-PCR as qRT-PCR

Line 100 – the same correction as above.

Line 102: “2.5. iTRAQ (Isobaric Tag for Relative and Absolute Quantitation) of proteins” The term quantification may be more appropriate here.

  • Response: thank you, Reviewer, we have revised it to reflect Quantification.

Figure 3

Notes on the parts A and B of the should be placed in similar corners of the figures (right or left)

  • Response: we have changed the orientation to the same side (right side justified).

Reviewer 2 Report

Authors performed proteomic study to assess the role of chromatin-modifying enzyme PHF8 in regulating the expression of PD-related synaptic plasticity proteins; SNCA, DNAJC6 (auxilin), SYNJ1 (synaptojanin 1), and SH3GL2 (endophilin A1). They assumed the possibility of targeting PHF8 as a potential disease-modifying therapeutic drug target in PD because of downregulation of synaptic plasticity proteins by PHF8 knockdown.

This assumption is not accurate because based on in vitro data only, authors can’t assume that PHF8 could be a disease-modifying therapy target in PD. To suppose that there is a need for in vivo study in PD animal model to show different neuroprotective mechanisms. Authors need to restate the conclusion in the abstract and summary.

Authors also need to add more information and restate some provided information in different sections of the manuscript.

Abstract

Please mention the abbreviation of any expression for the first time in the abstract then use the abbreviation in the remaining of abstract.

Introduction

Please add a few sentences from literature about the role of CAMKIIa, NCAM1, DNAJC6 (auxilin), SYNJ1 (synaptojanin 1), and SH3GL2 (endophilin A1) in synaptic functions and PD pathogenesis.

Please remove the repeated sentences. Line 36: “Through RNAi-based knockdown coupled with mass spectrometry-based quantitative proteomics, here we show that PHF8 is involved in upregulating key synaptic plasticity-related proteins in a neuronal activity-dependent manner”. This information is already mentioned in the beginning of third paragraph of introduction.

Table 1: Please move it to the result section and add p values.

Please mention the abbreviation of any expression once in the beginning of introduction section then use the abbreviation in the following sections.

Materials and Methods

Please add references to any protocol published before such as “Primary culture of cortical neurons” and “Immunofluorescence”

-       Cell lysate preparation and RNA / protein extraction:

Is there cDNA synthesis step in this protocol?

Did you use SYBR green in the qPCR assay? If so, please add these details in the method section.

Line 94: Please correct “rt-PCR”, it should be qPCR or qRT-PCR.

-       iTRAQ (Isobaric Tag for Relative and Absolute Quantitation) of proteins:

The abbreviation usually comes after the detailed name.

Page 5: This figure has to be mentioned in the text and has to have figure legend as well.

Can you explain to me why did you mix both control and test samples? How could you compare the expression of proteins in different samples after mixing?

Where is the section of statistical analysis in this study? Which tests have been used to compare the groups?

Results

All figures, tables, and supplementary tables have to be mentioned in order in the text. All figures should have a title then brief explanation of different panels of the figure.

-       Relative quantitation of the activity-regulated cortical neuronal proteome: Authors should mention in the text which supplementary table has the 2678 unique proteins. Please add p values in supplementary table 1.

Figure 1. Please restate the title, it is not accurate. The figure represents schematic representation of proteomic study protocol and normal distribution of both replicates (not data analysis pathway). Also please add letters (A, B, C, D) on the different panels of figure and explain this in the figure legend.

Please add the label “Biological Replicate 2” on the middle panel, it is missing.

-       RNA interference specifically depleted PHF8 in primary rat cortical neurons:

Where are the values in the text and is the difference between groups significant or no (please add p values).

Figure 2. The figure and its statistics need to be repeated. Expression of genes of interest should be normalized to the housekeeping gene (GAPDH) and represented as fold change in test group compared to control group. Significance level should be calculated and added to the figure. What is the title of Y-axis? Please make the cut on Y-axis to show value larger than 12100% on the figure. Please mention that “Data are represented as mean ± SEM.

-       Validation of Alpha-synuclein as a synaptic protein regulated by PHF8:

Figure 3: Please add a title. Can you clarify (In Figure 3A) how the overexpression of PHF8 does not ostentatiously affect Alpha-synuclein levels (How this is seen in the figure?). Similarly in Figure 3B, how the transfection with a shRNA plasmid against PHF8 seems to qualitatively reduce SNCA (How this is seen in the figure?).

-       Pathway analysis using DAVID reveals specific downregulation of proteins in PHF8 knockdown that are involved in synaptic function:

Can you clarify how you determined the 33 proteins involved in synaptic function?

Please provide the spreadsheet of DAVID analysis which has all the pathways enriched.

Discussion

Authors didn’t compare their results against any other published data in literature about the role of PHF8/its knockdown in PD pathogenesis. If no similar published studies, they may clarify that this is the first study to assess the role of PHF8 in in vitro model related to PD pathogenesis.

The manuscript needs English editing.

Author Response

Authors performed proteomic study to assess the role of chromatin-modifying enzyme PHF8 in regulating the expression of PD-related synaptic plasticity proteins; SNCA, DNAJC6 (auxilin), SYNJ1 (synaptojanin 1), and SH3GL2 (endophilin A1). They assumed the possibility of targeting PHF8 as a potential disease-modifying therapeutic drug target in PD because of downregulation of synaptic plasticity proteins by PHF8 knockdown.

This assumption is not accurate because based on in vitro data only, authors can’t assume that PHF8 could be a disease-modifying therapy target in PD. To suppose that there is a need for in vivo study in PD animal model to show different neuroprotective mechanisms. Authors need to restate the conclusion in the abstract and summary.

  • Response: thank you, Reviewer for this excellent correction. We have thus revised the phrasing of the Abstract to read: Given that the depletion of even a single chromatin-modifying enzyme can affect synaptic protein expression in such a concerted manner, more in-depth studies will be needed to show whether such a mechanism can be exploited as a potential disease-modifying therapeutic drug target in PD.

Authors also need to add more information and restate some provided information in different sections of the manuscript.

  • Response: Thank you. We have also revised the Discussion section to read:

4.2 Knockdown of PHF8 reduces levels of Alpha-synuclein

This proteomics study revealed that levels of SNCA reduced significantly with PHF8 knockdown, which we attempted to validateas then validated using immunofluorescence microscopy (Figure 3). The effective reduction of SNCA levels by PHF8 knockdown may raises the possibility of potentially using this mechanism as a potential therapeutic target by reducing the function of PHF8, either via gene therapy [65], or through pharmacological means, since PHF8 is an enzyme.

Abstract

Please mention the abbreviation of any expression for the first time in the abstract then use the abbreviation in the remaining of abstract.

  • Response: thank you for this kind reminder. We have now revised the abstract to have all abbreviations listed upfront.

Introduction

Please add a few sentences from literature about the role of CAMKIIa, NCAM1, DNAJC6 (auxilin), SYNJ1 (synaptojanin 1), and SH3GL2 (endophilin A1) in synaptic functions and PD pathogenesis.

  • Response: we have added the following sentences with the appropriate references:

Please remove the repeated sentences. Line 36: “Through RNAi-based knockdown coupled with mass spectrometry-based quantitative proteomics, here we show that PHF8 is involved in upregulating key synaptic plasticity-related proteins in a neuronal activity-dependent manner”. This information is already mentioned in the beginning of third paragraph of introduction.

  • Response: Thank you, Reviewer. We have removed the repeated sentence.

Table 1: Please move it to the result section and add p values.

  • Response: we have included the coefficient of variation (CV) values and revised the table accordingly.

Please mention the abbreviation of any expression once in the beginning of introduction section then use the abbreviation in the following sections.

  • Response: We have listed the abbreviations upfront.

Materials and Methods

Please add references to any protocol published before such as “Primary culture of cortical neurons” and “Immunofluorescence”

  • Response: we have added references to protocols that are already published (reference 5)

-       Cell lysate preparation and RNA / protein extraction:

Is there cDNA synthesis step in this protocol?

  • Response: yes, we have since added the following into the sentence: DNA was purified using a spin column (Qiagen), and eluted into 30 μl volumes; subsequently, 2 μl was used in qRT-PCR using primers against the transcriptional start site (TSS) of…

Did you use SYBR green in the qPCR assay? If so, please add these details in the method section.

  • Response: yes, we have now included this in the Methods section.

Line 94: Please correct “rt-PCR”, it should be qPCR or qRT-PCR.

  • Response: thank you, this has now been corrected to reflect qRT-PCR.

-       iTRAQ (Isobaric Tag for Relative and Absolute Quantitation) of proteins:

The abbreviation usually comes after the detailed name.

  • Response: thank you, we have corrected this so that the abbreviation comes after the detailed name.

Page 5: This figure has to be mentioned in the text and has to have figure legend as well.

Can you explain to me why did you mix both control and test samples? How could you compare the expression of proteins in different samples after mixing?

  • Response: the iTRAQ labelling reagent adds a distinct isobaric tag to control and test samples. Upon mixing the samples and running it via liquid chromatography – mass spectrometry, the mass spectrometer is programmed to do simultaneous and relative quantification of one isobaric tagged peptide to the other (since the peptides are labelled, the quantification is peptide-based, thus ensuring that the protein being quantified is the exact same protein but arising from two different samples)

Where is the section of statistical analysis in this study? Which tests have been used to compare the groups?

  • As we outlined in Figure 1, each peptide hit was obtained after running 2 different biological samples with each sample being run in duplicates and hence the ratio obtained via iTRAQ has a coefficient of variation (CV) value. This represents a confidence interval after averaging across 4 different runs.

Results

All figures, tables, and supplementary tables have to be mentioned in order in the text. All figures should have a title then brief explanation of different panels of the figure.

-       Relative quantitation of the activity-regulated cortical neuronal proteome: Authors should mention in the text which supplementary table has the 2678 unique proteins. Please add p values in supplementary table 1.

à Response: Thank you, Reviewer. We have now specifically referred to Supplementary Table 1 and added the Standard Deviation (SD) and Coefficient of Variation (CV) values into the Supplementary Table 1.

Figure 1. Please restate the title, it is not accurate. The figure represents schematic representation of proteomic study protocol and normal distribution of both replicates (not data analysis pathway). Also please add letters (A, B, C, D) on the different panels of figure and explain this in the figure legend.

  • Response: We have revised the title of Figure 1 to: Schematic representation of the proteomic study protocol and the geometric means of biological replicates used in the study. We have also included the following caption:
    Schematic representation of the proteomic study protocol and the geometric means of biological replicates used in the study. A) Workflow of proteomic experiment. B) Schematic of the iTRAQ quantitative proteomics protocol where two biological replicates are run through the mass spectrometer in duplicate (Run 1 and Run 2) which are then combined. C) Normogram of the number of peptides against the mean log2 frequency distribution.

Please add the label “Biological Replicate 2” on the middle panel, it is missing.

  • Response: Thank you. This has now been fixed.

-       RNA interference specifically depleted PHF8 in primary rat cortical neurons:

Where are the values in the text and is the difference between groups significant or no (please add p values).

  • Response: The p-value is <0.05; this has been reflected now in the revised figure caption.

Figure 2. The figure and its statistics need to be repeated. Expression of genes of interest should be normalized to the housekeeping gene (GAPDH) and represented as fold change in test group compared to control group. Significance level should be calculated and added to the figure. What is the title of Y-axis? Please make the cut on Y-axis to show value larger than 12100% on the figure. Please mention that “Data are represented as mean ± SEM.

  • Response: As our preliminary data showed that GAPDH being a glial gene was not entirely unaffected by the treatment conditions, we decided to normalize all 2 -delta(delta CT) values to RPL19 (NCBI Gene ID: 81767) which is a ribosomal gene which was not affected by any of the treatment conditions. The Y-Axis is the fold change over control expressed in percentage (title omitted for formatting).

-       Validation of Alpha-synuclein as a synaptic protein regulated by PHF8:

Figure 3: Please add a title. Can you clarify (In Figure 3A) how the overexpression of PHF8 does not ostentatiously affect Alpha-synuclein levels (How this is seen in the figure?). Similarly in Figure 3B, how the transfection with a shRNA plasmid against PHF8 seems to qualitatively reduce SNCA (How this is seen in the figure?).

  • Response: We have added a title to Figure 3: Immunofluorescence microscopy of the qualitative effects of PHF8 overexpression vis-à-vis PHF8 knockdown. We have also revised the figure caption to reflect that when PHF8 is overexpressed as the fusion protein PHF8-YFP, there is no change in the red signal (SNCA levels) as the cell still continued to have some red signals within it which when combined with green forms yellow. However, when PHF8 was knocked down using an shRNA (panel B), the cell did not turn yellow, reflecting the possibility that SNCA (red signal) might have been reduced.

-       Pathway analysis using DAVID reveals specific downregulation of proteins in PHF8 knockdown that are involved in synaptic function:

Can you clarify how you determined the 33 proteins involved in synaptic function?

Please provide the spreadsheet of DAVID analysis which has all the pathways enriched.

  • Response: the 33 proteins were hand-picked based on physiological function and role in disease (now we have revised the text to reflect this) and displayed in Table 1. DAVID analysis was shown in its entirety in Table 2.

Discussion

Authors didn’t compare their results against any other published data in literature about the role of PHF8/its knockdown in PD pathogenesis. If no similar published studies, they may clarify that this is the first study to assess the role of PHF8 in in vitro model related to PD pathogenesis.

  • Response: Thank you, Reviewer 2, for this very insightful comment. No comparison was possible as this is indeed the first study to assess the role of PHF8 in an in-vitro model of PD pathogenesis. We have included this in the discussion.

The manuscript needs English editing.

  • Response: Thank you. The manuscript has been edited by a native English speaker (AMJV in the author list)

Round 2

Reviewer 2 Report

Authors addressed most of the comments in the first review, but there are few points which still need modifications.

Materials and Methods

Any manuscript usually has section of statistical analysis at the end of materials and methods section in which different statistical tests used in the study are mentioned and the value at which the data considered significant such as p < 0.05. Please add this section at the end of Materials and Methods.

Results

The new version of manuscript still has the old version of Figure 1. Please correct that.

- Validation of Alpha-synuclein as a synaptic protein regulated by PHF8:

Figure 3 is not present in the text, please add it.

- Pathway analysis using DAVID reveals specific downregulation of proteins in PHF8 knockdown that are involved in synaptic function:

Table 2 looks the same as the first version of manuscript, please correct that.

Discussion

I don’t see the modified part (it is not highlighted in the text nor in different font color).

Author Response

Materials and Methods

Any manuscript usually has section of statistical analysis at the end of materials and methods section in which different statistical tests used in the study are mentioned and the value at which the data considered significant such as p < 0.05. Please add this section at the end of Materials and Methods.

--> RESPONSE: We have added this section at the end of Materials and Methods. 

cDNA was synthesized and subsequently purified using a spin column (Qiagen), and eluted into 30 μl volumes; thereafter, 2 μl was used in qRT-PCR employing SYBR Green  using primers against the transcriptional start site (TSS) of known activity-regulated neuronal genes such as Arc (NCBI Gene ID: 54323), BDNF (NCBI Gene ID: 24225), Fos (NCBI Gene ID: 314322), and control genes included Rpl19 (NCBI Gene ID: 81767) and GAPDH (NCBI Gene ID: 24383). Primers used for PHF8 qRT-PCR were: CCTAAAGCCCGTGTGACT and GGCGCGGCTGTTCTACCT. Statistical analyses were done using student’s two-tailed t-test with a p-value <0.05 being considered significant.

Results

The new version of manuscript still has the old version of Figure 1. Please correct that.

--> RESPONSE: apologies, we have now inserted the new (uploaded) Figure 1.

- Validation of Alpha-synuclein as a synaptic protein regulated by PHF8:

Figure 3 is not present in the text, please add it.

--> RESPONSE: apologies, we have included the new version of Figure 3 with its revised caption.

- Pathway analysis using DAVID reveals specific downregulation of proteins in PHF8 knockdown that are involved in synaptic function:

Table 2 looks the same as the first version of manuscript, please correct that.

--> RESPONSE: We have included the revised Table 2 with the top 30 ranked biological processes on DAVID. The full list has been uploaded as Supplementary Table 3.

Discussion

I don’t see the modified part (it is not highlighted in the text nor in different font color).

--> RESPONSE: We have revised the Discussion to include:

As this was the first attempt to assess the role of PHF8 in an in-vitro model of PD pathogenesis, we were not able to compare our results against published data in the literature. Nonetheless, the protein targets curated in the current dataset can be crossmatched with published PHF8 genetic targets obtained by various groups via ChipSeq (Supplementary Table 2; [72]).